# Investigation on the influence of airflow on the acoustic performance of a single–Cylinder diesel engine exhaust muffler

**Jun Fu** [1,2]*, **Milan Cheng**[1], **Yi Ma**[1], **Wei Zheng**[1]

**1** College of Mechanical and Energy Engineering, Shaoyang University, Shaoyang, China, **2** Key Laboratory of Hunan Province for Efficient Power System and Intelligent Manufacturing, Shaoyang University, Shaoyang, China

* 4160@hnsyu.edu.cn

## Abstract

The airflow in the exhaust muffler can affect the sound propagation characteristics. In this work, For a single-cylinder diesel engine, an experimental bench was set up, and the accuracy of the simulation model was verified through the mutual comparison between the experimental data and the simulation results. the transmission loss of exhaust muffler of a single-cylinder diesel engine is analyzed numerically. In addition, the influence of airflow on the acoustic performance of muffler is studied and analyzed in detail. Finally, a finite element method, namely automatic matched layer (AML) is used to simulate the anechoic boundary conditions, and the value of transmission loss of muffler with and without airflow is calculated. Results demonstrate that: Considering the influence of airflow, the transmission loss value of the muffler shows the obvious increase in the low-frequency domain of 0–2000 Hz, particularly below 1000 Hz, with difference up to 50 dB and an average of approx. 30dB. Nevertheless, the airflow has minimal influence on transmission loss in the medium-high frequency of 2000–7000 Hz. The acoustic performance is greatly affected by the internal fluid flow, and the fluid flow is beneficial to improving the acoustic performance of mufflers, especially in the low-frequency domain. Additionally, the change of transmission loss curve is more complex when the influence of airflow is considered. The research results of this work provide a more accurate prediction method on the transmission loss in practical application, and of reference significance for future studies on the acoustic performance of mufflers.

## 1. Introduction

Due to its sound fuel economy and durability, diesel engines have been widely adopted in transportation and agriculture [1,2]. Meanwhile, single-cylinder diesel engines dominate as the power source for industrial and agricultural purposes in developing countries [3,4]. Consequently, exhaust emission has caused considerable damage to the natural environment and human health [5]. Also, awareness is growing regarding the noise of single-cylinder diesel engines, which can be reduced sufficiently by using a properly designed muffler [6,7]. In addition, transmission loss is an important parameter for analyzing the

**Data availability statement:** All relevant data are within the paper and its Supporting information files.

**Funding:** This study was supported by the following funders: 1. Hunan Provincial Natural Science Foundation of China [Grant number 2022JJ50025] 2. Hunan Provincial Natural Science Foundation of China [Grant number 2023JJ50262] 3. 2024 Shaoyang University Scientific Research Special Project[Grant number 24KYQD16] 4. Postgraduate Scientific Research Innovation Project of Hunan Province [Grant number CX20231298] The funders had no role in study design, data collection and analysis, decision to publish, or preparation of the manuscript.

**Competing interests:** The authors have declared that no competing interests exist.

**Abbreviations:** $L$, The total length of the cavity (mm); $\delta_1$, The wall thickness of the intake pipe (mm); $D$, The diameter of the cavity (mm); $\delta_2$, The wall thickness of the partition pipe (mm); $D_1$, The diameter of the intake pipe (mm); $\delta_3$, The wall thickness of the cavity (mm); $D_2$, The diameter of the exhaust pipe (mm); $\delta_4$, The wall thickness of the exhaust pipe (mm); $D_3$, The diameter of the intake silencing hole (mm); $k$, The fluid heat transfer coefficient; $D_4$, Diameter of the exhaust silencing hole (mm); TL, Transmission loss; $D_5$, Diameter of the partition silencing hole (mm); $C_p$, Specific heat capacity; $S_\tau$, The internal heat source of the fluid; $\mu$, The dynamic viscosity; AML, automatic matched layer.

acoustic performance of the muffler [8]. Extensive theoretical and experimental studies have been conducted on transmission loss. Zhao [9] found that the compliant membrane motion gives rise to the production of transmission loss peaks. Magliacano et al. [10] proposed solutions for investigating the sound transmission loss of a typical fuselage panel section. Base on a wave and finite element method, Yang et al. [11] proposed a modeling strategy to predict transmission loss of multi-layered panels with fluid layers. Besides, the value of transmission loss is improved via a modified design of Helmholtz [12,13] and a topology-optimization-based design method [14]. The exhaust noise, which falls into the category of low-frequency noise, is the dominant noise source of a diesel engine [15]. The control of low-frequency has been studied by many researchers in the past and is of great current interest. A theoretical model was developed to study the influence of external mean flow on the sound transmission and unveiled that external mean flow has significant effects on the sound transmission loss in the low-frequency range [16]. Zhao et al. [17] changed the traditional structure of micro-perforated plate to enhance sound absorption performance at low frequencies. During the practical application of muffler, airflow is always present within. Therefore, the influence of airflow on acoustic characteristics should be investigated in detail. Based on Lighthill [18], Curle [19] and Ffowcs-Williams [20] made some researches and derived the Curle equation and the FW-H equation. The mean flow also has an obvious influence on sound propagation [21] and acoustic performance [22]. Consequently, scholars have carried out in-depth research on the mean flow. Jena et al. [23] demonstrated that the three-pole measurement method is no longer suitable for the calculation of transmission loss of muffler with mean flow. Hann et al. [24] evaluated transmission loss with the mean flow by the sine sweep method. Furthermore, Zhou et al. [25] examined the effect of Mach number of the external flow on sound transmission over a wide frequency range. Ji et al. [26] studied the influence of mean flow on the acoustic attenuation performance of straight-through perforated tube reactive silencers.

In the studies, the influence of external flow on sound transmission has been taken into consideration, yet for exhaust muffler of single-cylinder diesel engines, the influence of airflow on transmission loss had rarely been studied. Therefore, the focus of this work is the transmission loss of single-cylinder diesel engines exhaust mufflers with and without airflow. The accuracy of the model was verified through the mutual comparison between the experimental data and the simulation results.The transmission loss of exhaust muffler of the single cylinder diesel engine is analyzed numerically and the values of transmission loss of muffler with and without airflow are compared in this work.

## 2. Structural model and meshing

### 2.1. Structural size

The structural dimensions of the exhaust muffler of the single-cylinder diesel engine are shown in Table 1 and Fig 1. The main part of the muffler includes an intake pipe with several muffler holes, an expansion cavity, a perforated baffle, and an exhaust pipe with several muffler holes.

**Table 1. Basic parameters of the exhaust muffler.**

| Parameters | L | L1 | L2 | L3 | D | D1 | D2 | D3 | D4 | D5 | δ1 | δ2 | δ3 | δ4 |
|---|---|---|---|---|---|---|---|---|---|---|---|---|---|---|
| Values (mm) | 140 | 40 | 20 | 50 | 80 | 30 | 25 | 10 | 10 | 10 | 1 | 1 | 1 | 1 |

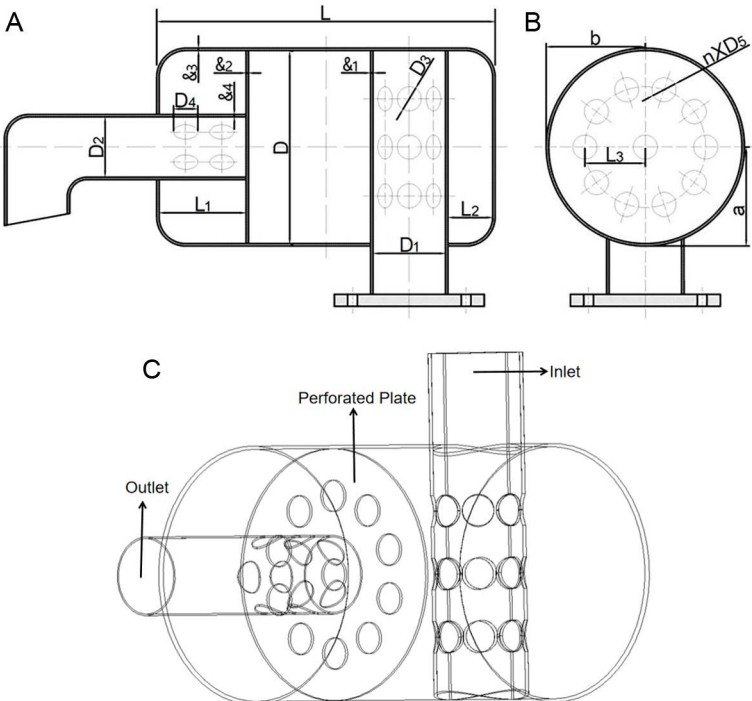

**Fig 1. Structure diagram of the exhaust muffler.** (a) Front view of the device (b) Cross-sectional view of the device (c) Three-dimensional model diagram of the device.

## 2.2. Grid division

Due to the excessive perforation in the intubation tube and partition, the internal structure of the muffler is complicated. Therefore, we chose to use the ICEM software to generate unstructured meshes for discretizing the model.The global grid size is set to 5.0 mm; the inlet and outlet size are set to 3.5 mm; the inlet wall and outlet wall size are set to 4.0 mm; the size of perforate hole in the entire cavity is set to 0.35 mm. The internal fluid domain after generating the grid is shown in Fig 2. The total number of mesh units is 1,673,938 and the total nodes are 291,638, which meets the accuracy requirements by grid independence verification. Reasonable grid size and step length are beneficial for reducing the time required for simulation while ensuring the accuracy of data. Figs 3 and 4 respectively display the grid independence verification and time independence verification.

## 3. Acoustic performance and flow field analysis of muffler

### 3.1. Basic theory of internal flow field of the muffler

The duct and cavity of exhaust muffler are mainly air fluid domains. Besides, the internal gas has a high flow velocity and the range of change is large, which makes the internal flow field distribution of muffler complicated. Nevertheless, air as a part of the fluid, it follows three basic conservation laws.

Law of conservation of mass:

$$\frac{\partial \rho}{\partial t} + \frac{\partial(\rho u)}{\partial x} + \frac{\partial(\rho v)}{\partial y} + \frac{\partial(\rho w)}{\partial z} = 0 \tag{1}$$

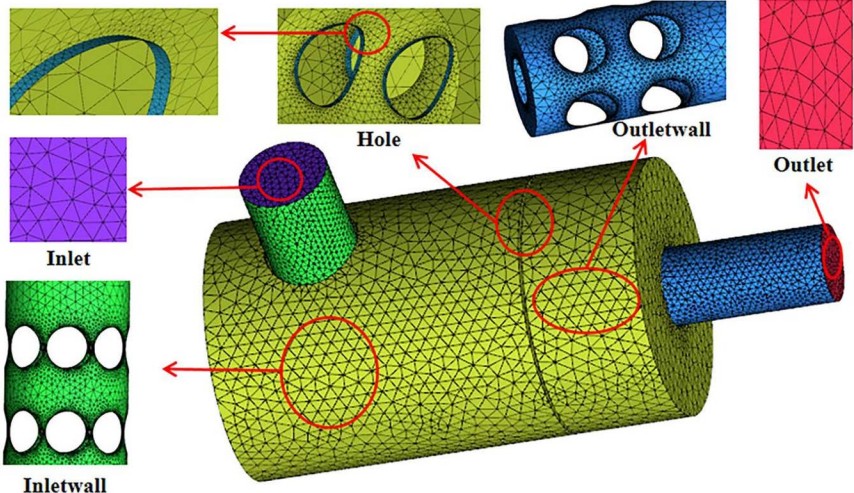

**Fig 2. The mesh diagram of muffler fluid analysis.**

Where $u$, $v$, and $w$ are the components of the velocity in the direction of X, Y, and Z respectively.

Law of conservation of momentum:

Expressed in the X direction as:

$$\frac{\partial(\rho u)}{\partial t} + \frac{\partial(\rho uu)}{\partial x} + \frac{\partial(\rho uv)}{\partial y} + \frac{\partial(\rho uv)}{\partial z}$$
$$= -\frac{\partial p}{\partial x} + \frac{\partial}{\partial x}\left(\mu\frac{\partial u}{\partial x}\right) + \frac{\partial}{\partial y}\left(\mu\frac{\partial u}{\partial y}\right) + \frac{\partial}{\partial z}\left(\mu\frac{\partial u}{\partial z}\right) + S_u \tag{2}$$

Expressed in the Y direction as:

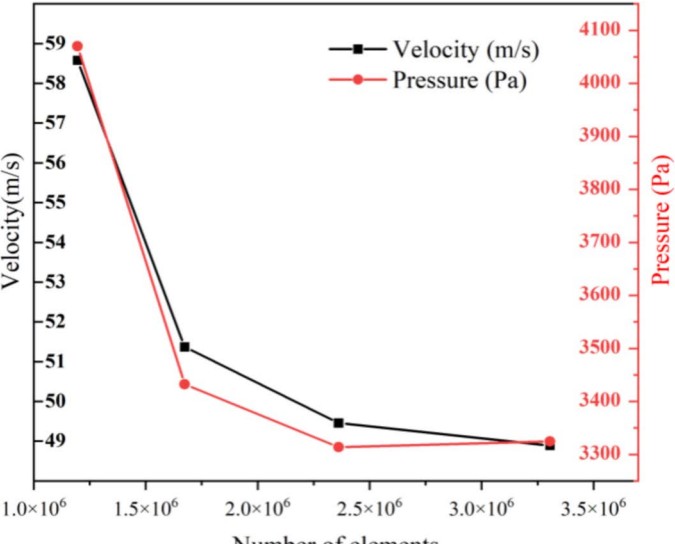

**Fig 3. Grid independence.**

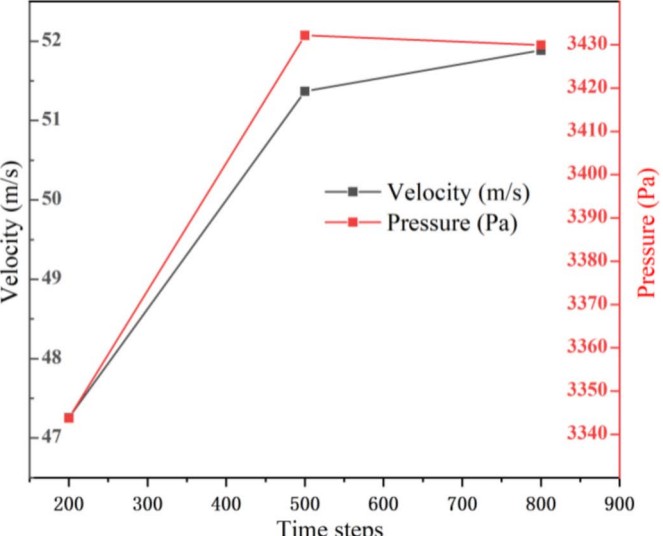

**Fig 4. Step-length independence.**

$$\frac{\partial(\rho v)}{\partial t} + \frac{\partial(\rho vu)}{\partial x} + \frac{\partial(\rho vv)}{\partial y} + \frac{\partial(\rho vw)}{\partial z}$$
$$= -\frac{\partial p}{\partial y} + \frac{\partial}{x}\left(\mu\frac{\partial v}{\partial x}\right) + \frac{\partial}{y}\left(\mu\frac{\partial v}{\partial y}\right) + \frac{\partial}{z}\left(\mu\frac{\partial v}{\partial z}\right) + S_v \tag{3}$$

Expressed in the Z direction as:

$$\frac{\partial(\rho w)}{\partial t} + \frac{\partial(\rho wu)}{\partial x} + \frac{\partial(\rho wv)}{\partial y} + \frac{\partial(\rho wv)}{\partial z}$$
$$= -\frac{\partial p}{\partial z} + \frac{\partial}{x}\left(\mu\frac{\partial w}{\partial x}\right) + \frac{\partial}{y}\left(\mu\frac{\partial w}{\partial y}\right) + \frac{\partial}{z}\left(\mu\frac{\partial w}{\partial z}\right) + S_w \tag{4}$$

Where $P$ is pressure, $\mu$ is the dynamic viscosity, $S_u$、 $S_v$ and $S_w$ are generalized source terms.

Law of conservation of energy

$$\frac{\partial \rho T}{\partial t} + \frac{\partial(\rho uT)}{\partial x} + \frac{\partial(\rho vT)}{\partial y} + \frac{\partial(\rho wT)}{\partial z}$$
$$= \frac{\partial}{\partial x}\left(\frac{k}{c_p}\frac{\partial T}{\partial x}\right) + \frac{\partial}{\partial y}\left(\frac{k}{c_p}\frac{\partial T}{\partial y}\right) + \frac{\partial}{\partial z}\left(\frac{k}{c_p}\frac{\partial T}{\partial z}\right) + S_\tau \tag{5}$$

Where $k$ is the fluid heat transfer coefficient, $T$ is the temperature, $C_p$ is the specific heat capacity, $S_\tau$ is the internal heat source of fluid, i.e., $S_\tau$ is the part where the mechanical energy of the fluid is converted into heat energy due to the viscous effect.

## 3.2. The numerical model and boundary conditions

The steps of the finite element analysis in this paper are as follows: model building, meshing, setting of boundary conditions, selection of physical models, setting of temperature conditions, solution, and post-processing of data. The specific flow chart is shown in Fig 5.

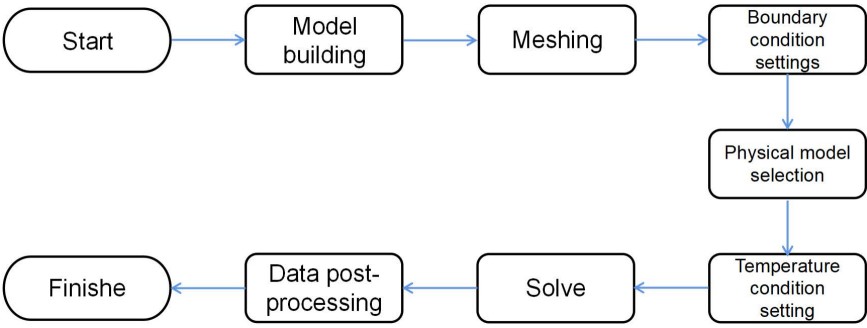

**Fig 5. Flow chart of finite element analysis.**

The inlet and outlet cross–sections of single–cylinder diesel engine exhaust muffler are small, with sound waves propagating mainly in the form of plane waves. For the numerical simulation in ANSYS Fluent, the outlet in the model is defined as an anechoic boundary condition, and the following assumptions are made for the acoustic conditions:

(1) The muffler shell is a rigid wall, and sound waves are not radiated outward on the wall;

(2) The propagation medium is an ideal uniform gas, and the static pressure and density of the medium are constant;

(3) During the propagation process, the sound waves are in an adiabatic state and there is no heat exchange with the outside.

The sound field boundary conditions are shown in Fig 6. The finite element AML (Automatic Matched Layer) method is used to simulate the anechoic boundary conditions and calculate the transmission loss.

The outlet boundary condition of the muffler is set as an anechoic boundary, the wall is a rigid wall, and the inlet sound wave is a plane wave, as shown in Fig 7. According to the requirements of calculation model of the muffler, the standard equation model $k$-$\varepsilon$ can be adopted to solve. It can solve the problem of the internal flow field of muffler in practical applications. The equations for solving the turbulent kinetic energy $k$ and the dissipation rate $\varepsilon$ are as follows:

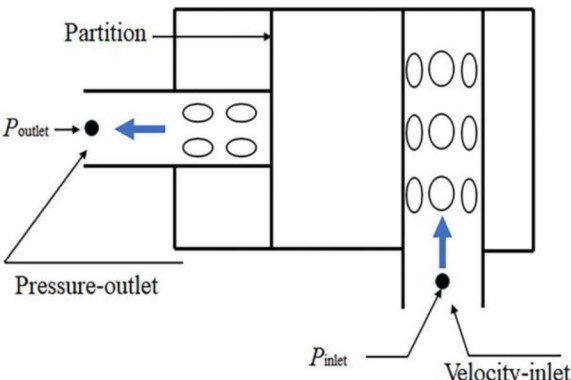

**Fig 6. Boundary condition of sound field.**

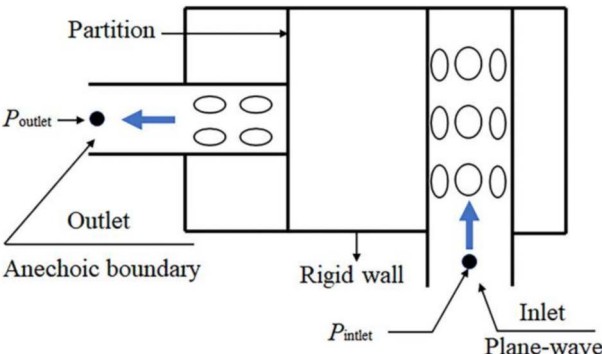

**Fig 7. Boundary condition of flow field.**

$$\rho \frac{Dk}{Dt} = \frac{\partial}{\partial x_i}\left[(\mu + \frac{\mu_t}{\delta_k})\frac{\partial k}{\partial x_i}\right] + G_k + G_b - \rho\varepsilon - Y_M \tag{6}$$

$$\rho \frac{D\varepsilon}{Dt} = \frac{\partial}{\partial x_i}\left[\left(\mu + \frac{\mu_t}{\delta_k}\right)\frac{\partial \varepsilon}{\partial x_i}\right] + G_{1\varepsilon}\frac{\varepsilon}{k}(G_k + G_{3\varepsilon} + G_b) - C_{2\varepsilon}\rho\frac{\varepsilon^2}{K} \tag{7}$$

Where, $C_{1\varepsilon} = 1.44$ $C_{2\varepsilon} = 1.92$ $C_\mu = 0.09$ $\sigma_k = 1.0$ $\sigma_\varepsilon = 1.3$ .

The influence of temperature has been taken into consideration in simulation. The temperature of muffler at the inlet is $556\,K$ and at the outlet is $450\,K$.

### 3.3. Test equipment and principle

Measurement of transmission loss imposes stringent requirements on experimental conditions. Given limited laboratory test bench equipment, this study approximates transmission loss through the method of insertion loss measurement. Table 2 lists the main equipment used for insertion loss experiments, including sound level meters, signal acquisition systems, and spectrum analyzers. Fig 8 illustrates the schematic diagram of the test bench setup.

For mufflers, the main evaluation indicators are transmission loss and insertion loss, which reflect the performance of mufflers from different aspects. However, due to the stringent experimental conditions required for the measurement of transmission loss and the high cost of equipment, this paper measures the insertion loss using the spatial five-point method due to the limitations of experimental conditions. Finally, the insertion loss is converted into an approximate value of the transmission loss according to the conversion formula (8).

$$\begin{cases} TL = L_{W_1} - L_{W_2} \\ IL = L_{P1} - L_{P2} \\ L_W = L_P + 20\lg r + 11 \end{cases} \tag{8}$$

**Table 2. Equipment list for noise measurement test system.**

| Name | Equipment Model | Name | Equipment Model | Name | Equipment Model |
|---|---|---|---|---|---|
| Monitoring and Control System | FC2000 | Sound Level Meter | HS5670B | High-performance computer | IBM 690 |
| Hydraulic dynamometer | GWD-160 | 1/3-Octave Band Filter | HS5731 | Exhaust Gas Analyzer | AVL DiGas4000 |
| Exhaust Back Pressure Controller | FC2050 | Power Sensor | N8262A | muffler | Self-made |

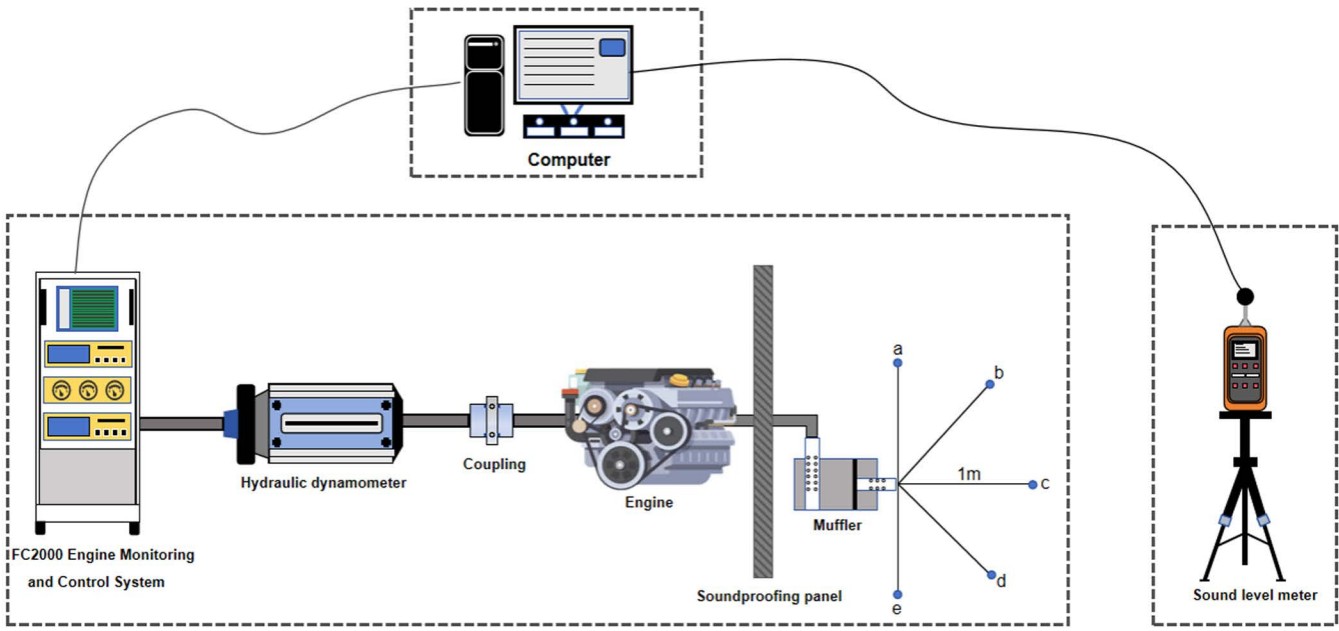

**Fig 8. Schematic diagram of the test bench.**

Where, $r$ is the distance from the tail end of the exhaust pipe to the test point ($r = 1$m in this paper), $L_{p1}$ and $L_{p2}$ are the sound pressure levels before and after the muffler is installed, $L_{w1}$ and $L_{w2}$ are the sound power levels at the entrance and exit respectively.

### 3.4. Data analysis

Combining the experimental and simulation data, the transmission loss 1/3 octave test and simulation diagrams were drawn. In Fig 9, the comparison between the test results of the 1/3 octave band-pass filter and the corresponding simulation results was presented. Since the minimum sampling frequency of the sound level meter used in the experiment was 20 Hz, 20 Hz was set as the starting point for comparison. As can be seen from the figure, within the frequency range from 20 Hz to 1250 Hz, the average difference between the two was approximately 2 dB, and the overall fitting was relatively good. However, a relatively large peak appeared near 1650 Hz, which was analyzed to be caused by the resonance frequency. Unfortunately, under the current experimental conditions, it was very difficult to conduct actual measurements at this frequency. Therefore, relatively obvious differences between the simulated values and the measured values appeared at this frequency. Overall, the fitting effect between the simulation results and the experimental results was relatively good, and the difference between the values was approximately 5%. Therefore, it is considered that the model and method proposed here are feasible.

### 3.5. Analysis of the sound field in the fluid domain

An input point and output point at the inlet and outlet of the muffler are defined respectively. According to the sound pressure response function curve, the change law of sound pressure response is analyzed. Without considering the influence of airflow, the sound pressure level frequency response function curve is shown in Fig 10.

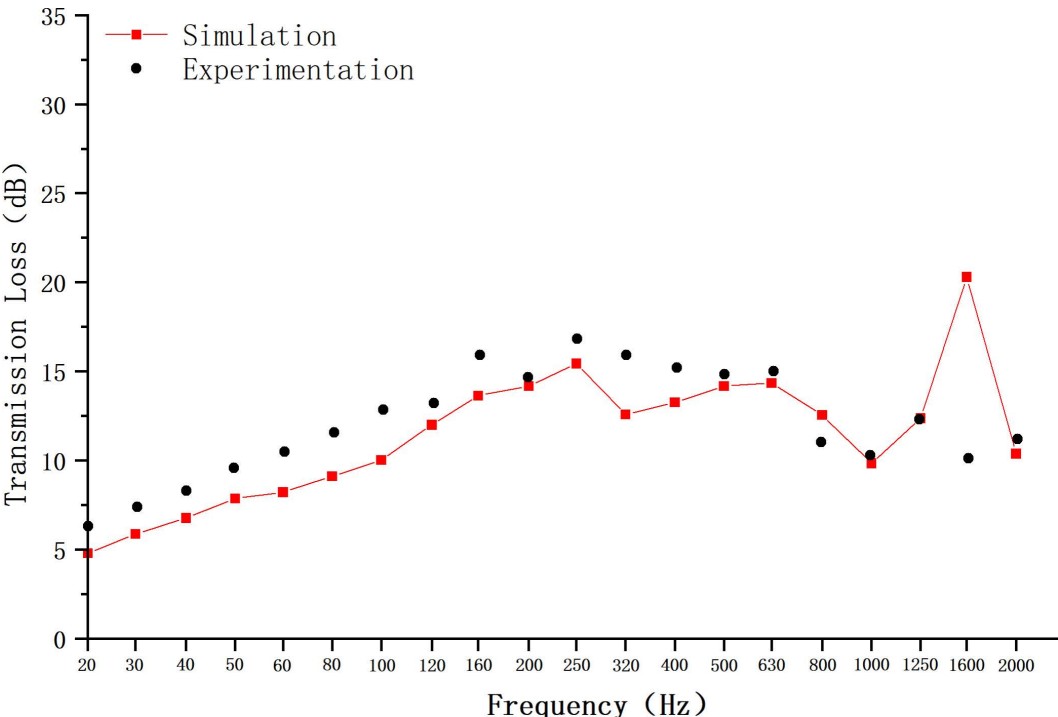

**Fig 9. Transmission loss 1/3 octave test and simulation diagram.**

The results in Fig 10 exhibit that: In the entire frequency range of 0–7000 Hz, the sound pressure at the outlet end of the muffler is less than that of the inlet, indicating that the muffler has a significant noise reduction effect. With further analysis, the sound pressure level change at the inlet end is not sharp in the low-frequency of 0–2000 Hz. However, the sound pressure level fluctuates greatly in the medium-high frequency of 2000–7000 Hz. Not only is the propagation mode of medium-high frequency sound waves more complicated, but also the sound wave reflections from the muffler wall, partition plate and cavity are increased. Thus, the range of change is significant. In the low-frequency range of 0–2000 Hz, the sound pressure response at the outlet end is also relatively flat. When the frequency exceeds 2000 Hz, the sound wave propagation model is complicated, and the sound wave reflections on the wall of muffler increase. Due to the uncertainty of the propagation direction of each sound wave, the change of the sound pressure transmitted into inlet and outlet is complex.

In the exhaust muffler, the fluid medium has a great influence on sound propagation. Therefore, the sound pressure cloud diagram of the muffler considering the influence of airflow is derived, as shown in Fig 11.

When the frequency is less than 260 Hz, the internal sound of the muffler is transmitted in a plane wave in the inlet pipe. The sound pressure level is basically unchanged after entering the cavity and transmitted in a uniform plane wave. The changes of sound pressure amplitude diagram with and without airflow influence demonstrate high similarity.

For further clarification, the sound pressure response function curve of the inlet and outlet is also calculated, as shown in Fig 12.

In the frequency range of 0–900 Hz, the sound pressure at the inlet of the muffler is significantly higher than that of the outlet, with an average of approx. 50 dB. According to the outlet sound pressure cloud diagram in the range of 0–2000 Hz, the airflow has

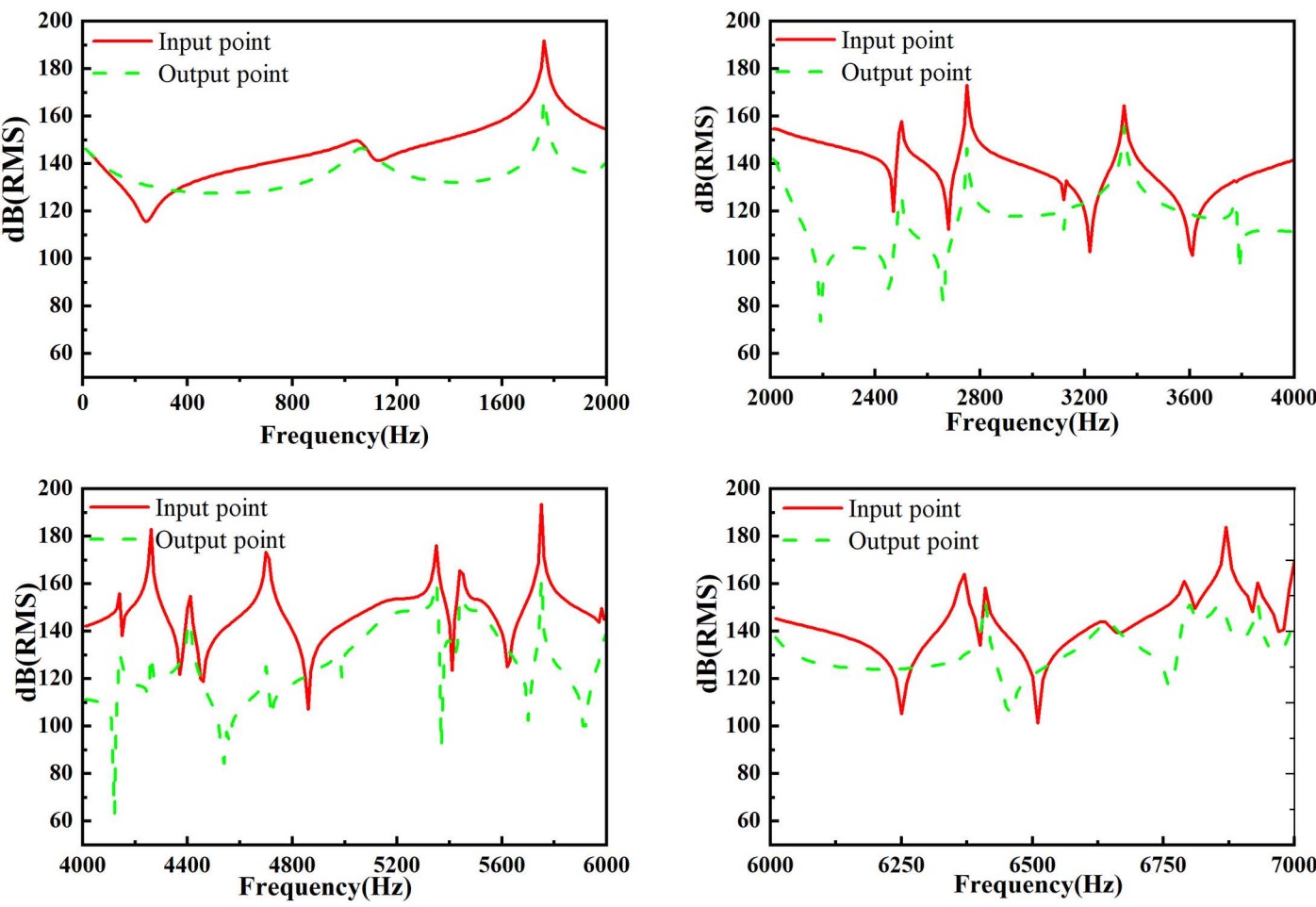

**Fig 10. The curve of sound pressure level without airflow.**

minimal effect on the sound pressure. Comparative analysis of Figs 10 and 12 shows that the sound pressure level changes little at the outlet in the range of 0–2000 Hz. Considering the airflow, the sound pressure level at the outlet end is significantly increased in the low-frequency domain of 0–2000 Hz. The airflow will affect the silencing performance of muffler, but it exerts a small effect on the sound pressure level at the inlet in medium-high frequency. As a matter of fact, the airflow of fluid increases the overall average sound pressure level, particularly at the inlet end. This shows that the sound pressure level at the inlet of the muffler is underestimated without considering the influence of airflow.

## 4. The influence of airflow on transmission loss

### 4.1. The transmission loss with airflow and without airflow

In practical applications of mufflers, airflow always presents in the cavity. Due to the internal structure of muffler, the airflow produces a pressure difference when passing through the muffler, which has a significant influence on the acoustic performance of the muffler. Therefore, it is necessary to probe further into the influence of airflow on acoustic performance. As one of the most-frequently-used indexes to evaluate the acoustic performance

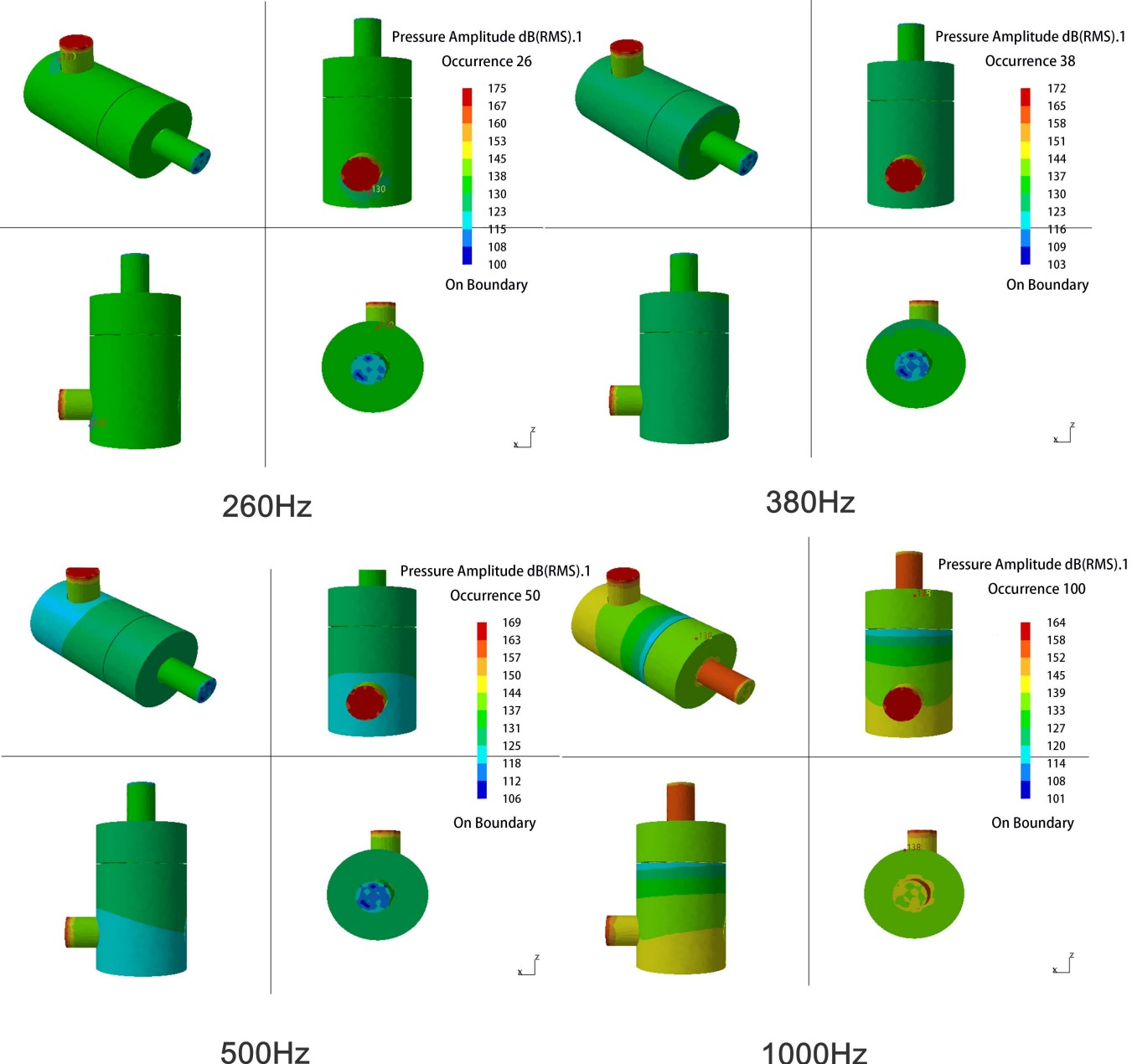

**Fig 11. Cloud diagram of the sound pressure level amplitude with the influence of airflow.**

of mufflers, transmission loss is an important indicator of the acoustic performance of mufflers. Therefore, this work proceeds with studies on the influence of airflow on transmission loss.

The calculation formula of the transmission loss is:

$$TL = 10\log_{10}\left(\frac{P_2 \times \overline{P_2}}{P_1 \times \overline{P_1}}\right) \tag{9}$$

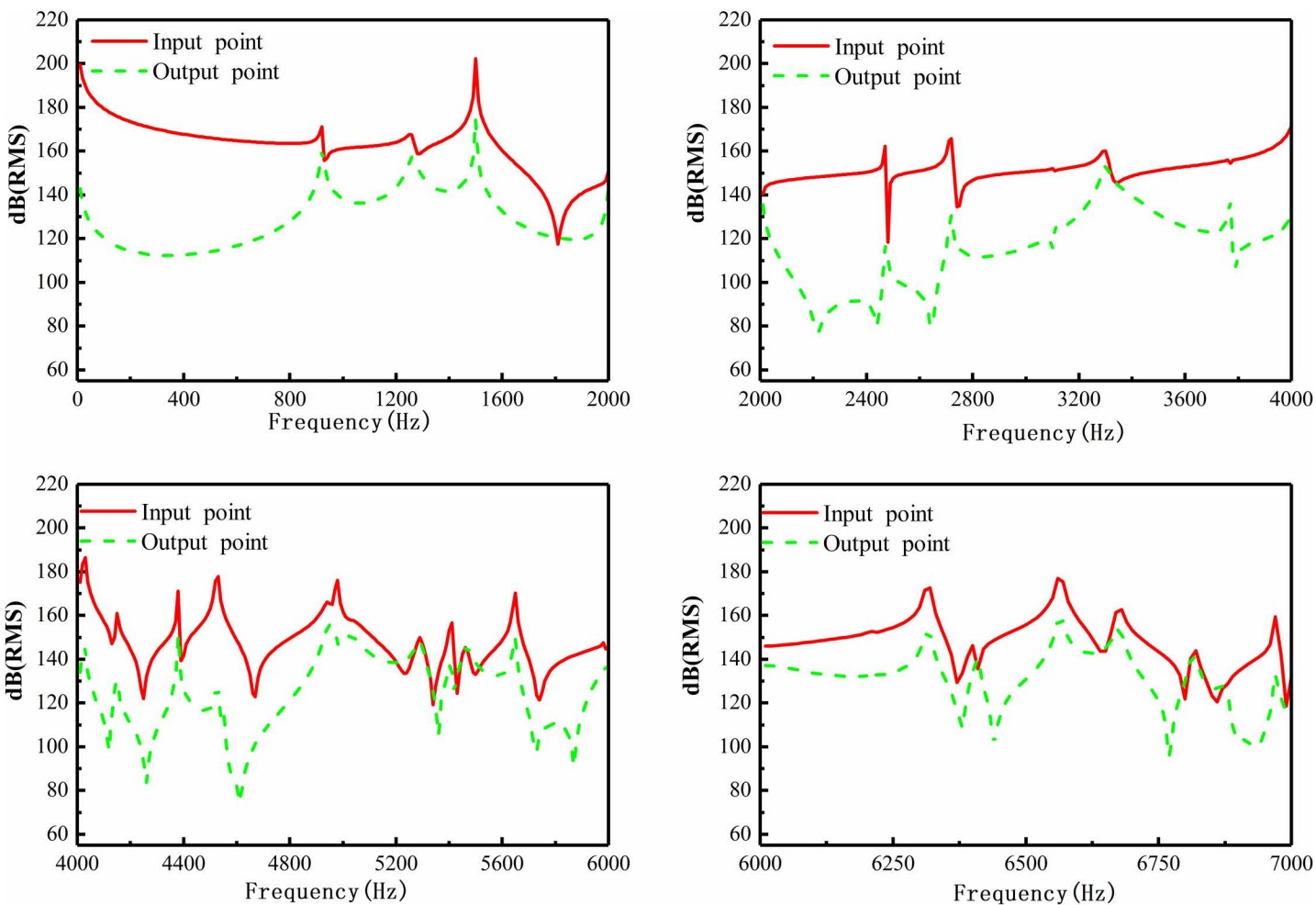

**Fig 12. The curve of sound pressure level with airflow.**

Where, $P_2 = (P_{inlet} + p_c)/2$; $P_1 = P_{out}$; $P_{inlet}$ is the sound pressure at the inlet, $p_c$ is the sound pressure of the surrounding environment and $P_{out}$ is the sound pressure at the outlet. $\overline{P}$ is a complex conjugate of sound pressure. According to the sound pressure response function of the input point and the output point of muffler, combined with Formula 9, the transmission loss of the muffler can be obtained, as shown in Fig 13. Meanwhile, the comparison chart of the transmission loss with and without airflow is also demonstrated in Fig 13.

In the frequency domain of 20–4500 Hz, the difference in transmission loss between the cases with and without airflow is approximately 11%.In the frequency domain I (20–2000 Hz), the value of transmission loss displays a considerable difference with or without airflow. The difference is up to 50 dB and the average is approx. 30 dB. In this frequency domain, the transmission loss curves with and without airflow hardly fit. The value of transmission loss is negative in points A and B without airflow, showing the muffler not only fails to reduce noise, but also becomes a noise amplifier. In the frequency III (20–1300 Hz), the value of transmission loss is very low, mostly below 10 dB. In the frequency range I (20–2000 Hz), the value of transmission loss is significantly higher than that without airflow. In addition, the transmission loss indicates no negative value in this frequency domain and the noise reduction effect is sound. According to the comparison chart, the transmission loss curve with and without airflow fits very well in frequency range II.

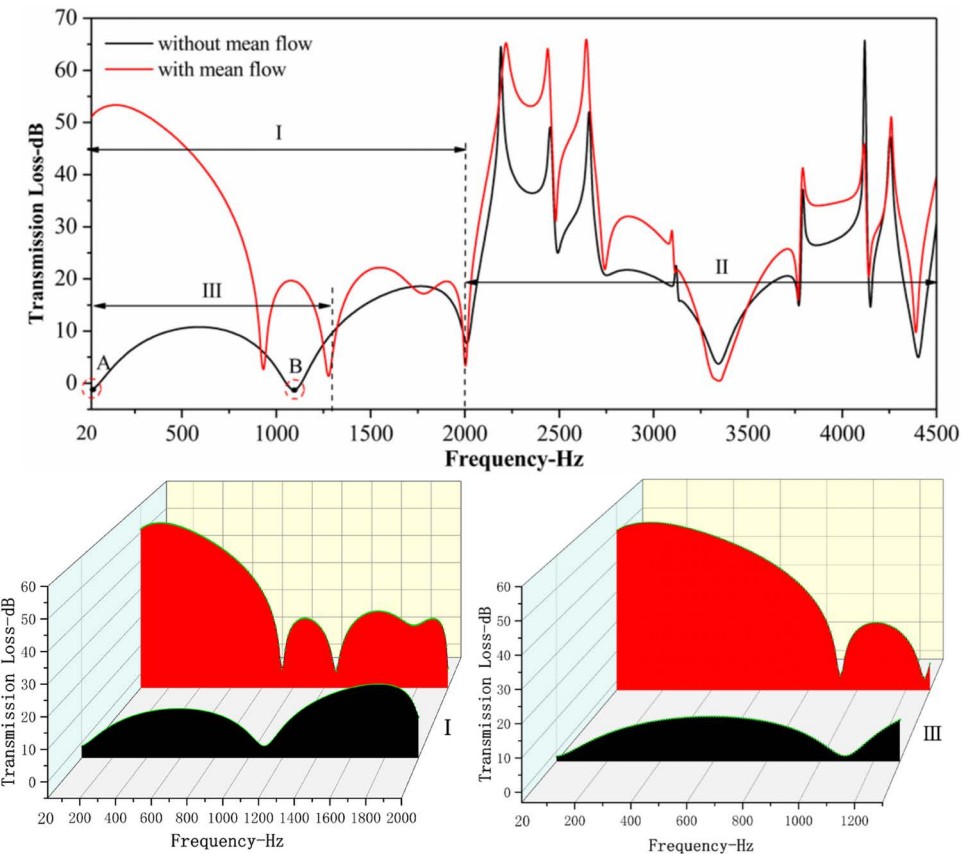

**Fig 13. The comparison chart of the transmission loss with airflow and without airflow.**

In consequence, airflow exerts a minimal effect on the acoustic performance of muffler in this frequency. Overall, airflow has a great influence on the acoustic performance of mufflers in the low-frequency range, while in the middle-high frequency range, the airflow of gas has little effect on the acoustic performance of muffler. This conclusion is consistent with the research of Liu [16].

## 4.2. Analysis of the airflow field of the muffler at different airflow rates

It is obvious that airflow can be of influence on the silencing performance of exhaust muffler. Consequently, the study of different airflow rates is of great significance in the study of the acoustic performance of mufflers. The velocity cloud diagram distribution with different exhaust airflows shows high similarity in Fig 14, indicating that the internal structure of muffler has same effect on the airflow velocity. The main difference is that the gradients of speed variation are different with different exhaust airflows.

At each exhaust velocity, the airflow first passes through the perforation on the inlet insertion pipe for split airflow, then airflow goes into the cavity when it flows through the first-stage perforation of the insertion pipe. As the volume of the chamber is greatly increased, the airflow velocity is greatly reduced. The remaining gas of the insertion tube continues to flow into the cavity through the second and third perforations into the tube. With a large quantity of gas in the original cavity, the airflow velocity at the second and third perforations decreases. It is obviously much smaller than the first-order perforation.

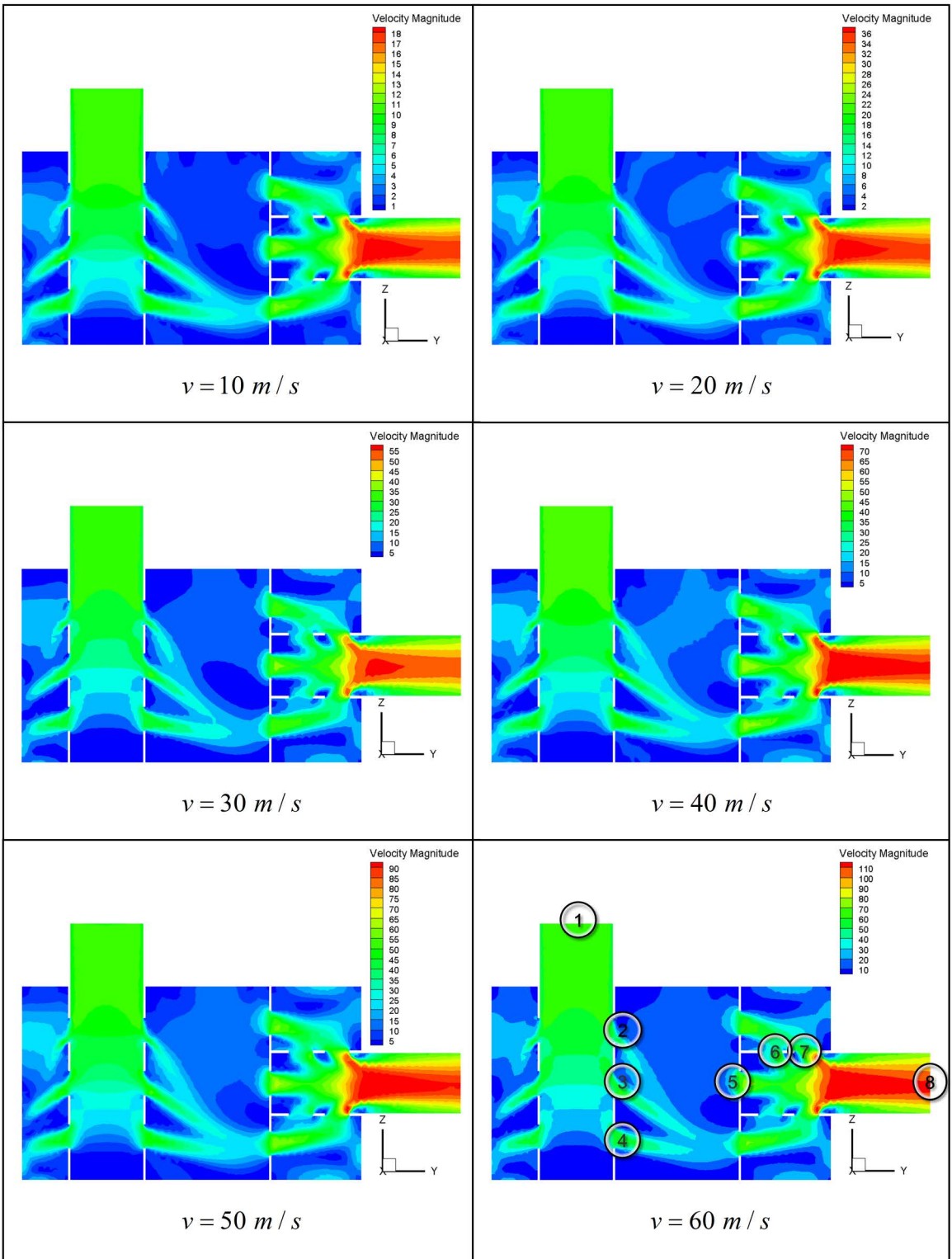

**Fig 14. The internal velocity cloud diagram of the muffler with different inlet airflow rates.**

Meanwhile, the velocity near the perforation is relatively high before the airflow reaches the perforation of partition. Due to the expansion of the volume in the chamber, the average air velocity in the cavity is reduced significantly comparared to the air velocity in the inlet and outlet pipes. In the meantime, the airflow becomes very gentle where the angle of the cavity changes considerably and forms a vortex, as dark blue areas demonstrated in Fig 14. After the airflow reaches the partition, the airflow speed increases due to the shrinkage of small holes. Part of the gas flows directly into the exhaust tailpipe through the central hole. The remaining airflows into the second cavity through other perforations, and then goes into the exhaust tailpipe through the perforation on the insertion pipe. The air velocity in the exhaust pipe increases again and finally flows out. As shown in Fig 14, the airflow velocity distribution is very uneven no matter in the cavity or the exhaust pipe.

To facilitate quantitative analysis of the speed distribution changes in muffler structure, 8 typical test points were selected, as shown in Fig 15. Airflow velocity values at each test point with different inlet flow velocities are also shown in Fig 15.

From Fig 15, the velocity of airflow through the outlet is much greater than the inlet. This is because the cross-section of the exhaust tailpipe is larger than that of the intake insertion pipe. According to the principle of momentum conservation, the total amount of gas flowing into the muffler cavity is equal to the amount flowing out of the exhaust tailpipe. In this work, the inlet diameter of muffler calculation model is 30 $mm$, and the diameter of the outlet is 25 $mm$. Therefore, the airflow velocity of the exhaust pipe should be greater than that of the inlet pipe. Meanwhile, test results in Fig 15 are also in line with the principle. According to the analysis of the velocity data of measured points, the maximum descending speed is the first-order perforation of the intake tube. From 10$m/s$ to 60$m/s$, the best rate of decrease is 10$m/s$. As the intake speed increases, the speed gradually decreases. When the speed is 50$m/s$, the speed drop rate reaches the lowest value of 31.08%. In all, the exhaust muffler can effectively reduce the airflow velocity.

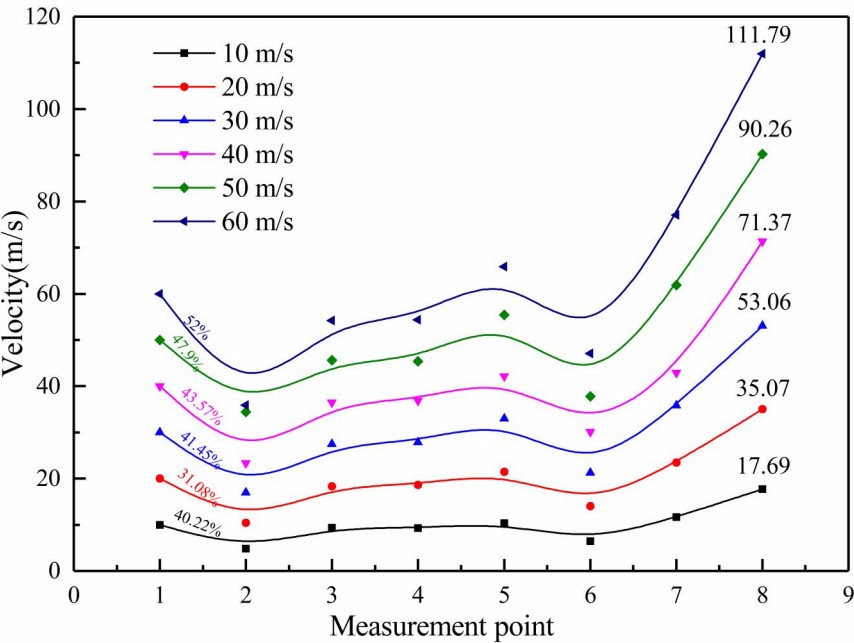

**Fig 15. Airflow velocity at different measurement points.**

## 5. Conclusions

This work investigates the influence of airflow on the transmission loss of the exhaust muffler by adopting the AML method. Based on the acoustic theory, the influence of the internal airflow field of the muffler and the wall vibration response of the muffler on the acoustic performance are taken into consideration comprehensively. Meanwhile, the sound pressure vibration response at the inlet and outlet, and the overall sound pressure amplitude change are analyzed. The comparison chart of the transmission loss with and without airflow is also obtained.

The research results show that: Considering the influence of the airflow, the transmission loss is increased by an average of 30 dB in the frequency range of 0–2000 Hz. It is verified that airflow exerts a great influence on transmission loss of the muffler in the low-frequency range. Nevertheless, the airflow has minimal influence on transmission loss in the medium-high frequency of 2000–7000 Hz. Without considering the influence of airflow on acoustic performance, the value of transmission loss is negative at 900 Hz and 1300 Hz. Under such circumstance, the muffler not only does not reduce noise but contribute to sound amplification instead. Therefore, the internal airflow of the diesel engine has a great effect on the suppression of exhaust noise.

During the practical application of muffler, airflow is always present within. Therefore, it is of practical significance to consider the influence of airflow on the acoustic performance of the muffler. This finding of this work can be helpful to assist in the implementation of the exhaust muffler of a single cylinder diesel engine in reducing unwanted engine noise and improving acoustic performance.

## Supporting information

**S1 Data. Abbreviation Comparison Table for the Full Text.**
(XLSX)

## Author contributions

**Conceptualization:** Jun Fu.

**Data curation:** Jun Fu.

**Formal analysis:** Jun Fu, Milan Cheng.

**Funding acquisition:** Jun Fu, Milan Cheng.

**Investigation:** Milan Cheng, Wei Zheng.

**Methodology:** Milan Cheng, Wei Zheng.

**Project administration:** Milan Cheng, Yi Ma, Wei Zheng.

**Resources:** Yi Ma, Wei Zheng.

**Software:** Yi Ma.

**Supervision:** Yi Ma.

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
