## [Decision Letter · Decision Letter 0]

26 Nov 2024

PONE-D-24-50090The effect of the airflow on the effect of the mufflerPLOS ONE

Dear Dr. Fu,

Thank you for submitting your manuscript to PLOS ONE. After careful consideration, we feel that it has merit but does not fully meet PLOS ONE’s publication criteria as it currently stands. Therefore, we invite you to submit a revised version of the manuscript that addresses the points raised during the review process.

In addition to the reviewers comments, the authors are suggested to compare their results with previous experimental studies and include comparison in the revised paper.

We look forward to receiving your revised manuscript.

Kind regards,

Muhammad Shakaib, PhD

Academic Editor

PLOS ONE

Journal requirements: When submitting your revision, we need you to address these additional requirements. 1. Please ensure that your manuscript meets PLOS ONE's style requirements, including those for file naming. The PLOS ONE style templates can be found at https://journals.plos.org/plosone/s/file?id=wjVg/PLOSOne_formatting_sample_main_body.pdf and https://journals.plos.org/plosone/s/file?id=ba62/PLOSOne_formatting_sample_title_authors_affiliations.pdf  2. Please include a caption for figure 4 and 6. 3. PLOS requires an ORCID iD for the corresponding author in Editorial Manager on papers submitted after December 6th, 2016. Please ensure that you have an ORCID iD and that it is validated in Editorial Manager. To do this, go to ‘Update my Information’ (in the upper left-hand corner of the main menu), and click on the Fetch/Validate link next to the ORCID field. This will take you to the ORCID site and allow you to create a new iD or authenticate a pre-existing iD in Editorial Manager. 4. We note that the grant information you provided in the ‘Funding Information’ and ‘Financial Disclosure’ sections do not match.  When you resubmit, please ensure that you provide the correct grant numbers for the awards you received for your study in the ‘Funding Information’ section. 5. Thank you for stating the following in the Acknowledgments Section of your manuscript: [This work was supported by Hunan Provincial Natural Science Foundation of China [Grant number 2022JJ58025], Hunan Provincial Natural Science Foundation of China [Grant number 2023JJ50262], 2024 Shaoyang University Scientific Research Special Project[Grant number 24KYQD16], Postgraduate Scientific Research Innovation Project of Hunan Province [Grant number CX20231298].]We note that you have provided funding information that is not currently declared in your Funding Statement. However, funding information should not appear in the Acknowledgments section or other areas of your manuscript. We will only publish funding information present in the Funding Statement section of the online submission form. Please remove any funding-related text from the manuscript and let us know how you would like to update your Funding Statement. Currently, your Funding Statement reads as follows:  [The author(s) received no specific funding for this work.] Please include your amended statements within your cover letter; we will change the online submission form on your behalf. 6. We note that you have indicated that there are restrictions to data sharing for this study. PLOS only allows data to be available upon request if there are legal or ethical restrictions on sharing data publicly. For more information on unacceptable data access restrictions, please see http://journals.plos.org/plosone/s/data-availability#loc-unacceptable-data-access-restrictions.  Before we proceed with your manuscript, please address the following prompts: a) If there are ethical or legal restrictions on sharing a de-identified data set, please explain them in detail (e.g., data contain potentially identifying or sensitive patient information, data are owned by a third-party organization, etc.) and who has imposed them (e.g., a Research Ethics Committee or Institutional Review Board, etc.). Please also provide contact information for a data access committee, ethics committee, or other institutional body to which data requests may be sent. b) If there are no restrictions, please upload the minimal anonymized data set necessary to replicate your study findings to a stable, public repository and provide us with the relevant URLs, DOIs, or accession numbers. For a list of recommended repositories, please seehttps://journals.plos.org/plosone/s/recommended-repositories. You also have the option of uploading the data as Supporting Information files, but we would recommend depositing data directly to a data repository if possible. We will update your Data Availability statement on your behalf to reflect the information you provide.

Reviewers' comments:

Reviewer's Responses to Questions

**Comments to the Author**

1. Is the manuscript technically sound, and do the data support the conclusions?

Reviewer #1: Yes

Reviewer #2: Partly

2. Has the statistical analysis been performed appropriately and rigorously? 

Reviewer #1: Yes

Reviewer #2: Yes

3. Have the authors made all data underlying the findings in their manuscript fully available?

Reviewer #1: Yes

Reviewer #2: Yes

4. Is the manuscript presented in an intelligible fashion and written in standard English?

Reviewer #1: Yes

Reviewer #2: No

5. Review Comments to the Author

Reviewer #1: Reviewer comments:

- How can the performance of muffler in other industrial applications such as power generators or heavy vehicles be improved using these findings?

- Does the effect of airflow on mufflers differ in diesel engines compared to gasoline engines or electric motors?

- What are the potential effects of airflow on fuel consumption and efficiency of a diesel engine?

- Are there additional factors such as airflow velocity or air density that significantly affect these results?

- Do operating conditions significantly affect the effectiveness of sound insulation?

- How can future studies in this area be improved to include other effects such as humidity or high temperatures on muffler performance?

- Please check and correct the symbols in Equations 2 and 4.

- Where P is pressure, μ is the dynamic viscosity, Su、 Sv and Sw are generalized source terms. In the above paragraph, check the symbols as in the equations.

- In Figures 3 and 4, the axes should be the same color as the axis titles.- In Figures 8 and 11, the color bars across all images are not clearly defined, making it difficult to interpret the data accurately.

- Check the equation 8 and identify some symbols.

Reviewer #2: In this research work, the authors address an interesting problem concerning the influence of airflow in mufflers. They state that the effect of airflow on transmission loss within mufflers has not been extensively studied. To bridge this gap, the authors adopt a numerical approach to analyze and present their findings. However, several observations can be made regarding the paper and its conclusions:

1. Structure and Writing Style: The paper follows a conventional manuscript structure, which is commendable. While the authors have made significant efforts in writing the paper, the cohesiveness of the text is lacking. Improving the overall flow and grammar would enhance readability for the audience.

2. Title Improvement: The title could be more self-explanatory to provide readers with a clearer idea of the paper's content.

3. Details on Numerical Analysis Tools: A critical omission in the paper is the lack of information about the type of numerical or simulation software used for the analysis. There is no mention of the software used for mesh generation, which is crucial for replication and validation.

4. Methodology Clarification: Including a flowchart outlining the Finite Element Analysis (FEA) process would significantly improve the reader's understanding of the methodology.

5. 3D CAD Model: In Section 2, presenting a 3D CAD model of the exhaust muffler with labeled components would aid in the comprehension of the design.

6. Figure Accuracy and Quality:

• There are spelling errors in Figure 7 that need correction.

• The overall quality of the figures should be significantly enhanced for better visual presentation.

7. Graphical Representation of Results: Can the authors graphically illustrate the change in transmission loss? Such visualizations would make the findings more accessible to readers.

8. Error Analysis and Validation: The authors should provide an estimation of the percentage error in transmission loss due to airflow conditions. Experimental validation to support these findings would add credibility to the work.

9. Further Findings and Experimental Evidence: To improve clarity, additional findings and, if feasible, experimental validation should be reported.

10. Significance and Context: Readers may question the significance of this work given that previous studies have addressed similar issues. The authors justify their analysis, but it is essential to highlight what distinguishes this work from prior studies. For instance, works by Hirata and Itow (Influence of Air Flow on the Attenuation Characteristics of Resonator Type Mufflers), Zhigang Chu (Effects of Airflow on the Acoustic Attenuation Performance of Reactive Mufflers), and Zeynep Parlar (Acoustic and Flow Field Analysis of a Perforated Muffler Design) have explored the impact of geometry, materials, and steady/unsteady airflow on mufflers. These studies, among others, discuss transmission loss in relation to airflow and other parameters. The current study should explicitly emphasize its novel contributions in this context.

6. PLOS authors have the option to publish the peer review history of their article (what does this mean? ). If published, this will include your full peer review and any attached files.

**Do you want your identity to be public for this peer review?** For information about this choice, including consent withdrawal, please see our Privacy Policy .

Reviewer #1: No

Reviewer #2: No

---

## [Author Response · Author response to Decision Letter 1]

5 Jan 2025

Dear reviewers and editor,

The reviewers' and editor's comments have been answered and replied to item by item in the attachment.

---

## [Decision Letter · Decision Letter 1]

12 Jan 2025

Investigation on the Influence of Airflow on the Acoustic Performance of a Single - Cylinder Diesel Engine Exhaust Muffler

PONE-D-24-50090R1

Dear Dr. Fu,

We’re pleased to inform you that your manuscript has been judged scientifically suitable for publication and will be formally accepted for publication once it meets all outstanding technical requirements.

Kind regards,

Muhammad Shakaib, PhD

Academic Editor

PLOS ONE

Additional Editor Comments (optional):

Reviewers' comments:

Reviewer's Responses to Questions

**Comments to the Author**

1. If the authors have adequately addressed your comments raised in a previous round of review and you feel that this manuscript is now acceptable for publication, you may indicate that here to bypass the “Comments to the Author” section, enter your conflict of interest statement in the “Confidential to Editor” section, and submit your "Accept" recommendation.

Reviewer #1: All comments have been addressed

Reviewer #2: All comments have been addressed

2. Is the manuscript technically sound, and do the data support the conclusions?

Reviewer #1: Yes

Reviewer #2: Yes

3. Has the statistical analysis been performed appropriately and rigorously? 

Reviewer #1: Yes

Reviewer #2: Yes

4. Have the authors made all data underlying the findings in their manuscript fully available?

Reviewer #1: (No Response)

Reviewer #2: Yes

5. Is the manuscript presented in an intelligible fashion and written in standard English?

Reviewer #1: (No Response)

Reviewer #2: Yes

6. Review Comments to the Author

Reviewer #1: They have made every effort to improve the manuscript and made some changes marked in red during the revision process. I sincerely appreciate the enthusiastic work of the authors.

Sincerely,

Ali

Reviewer #2: Thank you to the authors for revising the manuscript. It is now well designed and carries value. Please ensure that the paragraphs are arranged in a Justified style and correct the capitalizations.

7. PLOS authors have the option to publish the peer review history of their article (what does this mean? ). If published, this will include your full peer review and any attached files.

**Do you want your identity to be public for this peer review?** For information about this choice, including consent withdrawal, please see our Privacy Policy .

Reviewer #1: No

Reviewer #2: No

---

## [Editor Report · Acceptance letter]

PONE-D-24-50090R1

PLOS ONE

Dear Dr. Fu,

I'm pleased to inform you that your manuscript has been deemed suitable for publication in PLOS ONE. Congratulations! Your manuscript is now being handed over to our production team.

Kind regards,

on behalf of

Dr. Muhammad Shakaib

Academic Editor

PLOS ONE